# A Vision and Proof of Concept for New Approach to Monitoring for Safer Future Smart Transportation Systems

**DOI:** 10.3390/s24186018

**Published:** 2024-09-18

**Authors:** Kent X. Eng, Yang Xie, Mauricio Pereira, Zygmunt J. Haas, Samir R. Das, Petar M. Djurić, Branko Glisic, Milutin Stanaćević

**Affiliations:** 1Department of Civil and Environmental Engineering, Princeton University, Princeton, NJ 08544, USA; mp34@princeton.edu (M.P.); bglisic@princeton.edu (B.G.); 2Department of Electrical and Computer Engineering, Stony Brook University, Stony Brook, NY 11794, USA; yang.xie.2@stonybrook.edu (Y.X.); petar.djuric@stonybrook.edu (P.M.D.); milutin.stanacevic@stonybrook.edu (M.S.); 3Computer Science Department, University of Texas at Dallas, Richardson, TX 75080, USA; haas@utdallas.edu; 4Department of Computer Science, Stony Brook University, Stony Brook, NY 11794, USA; samir@cs.stonybrook.edu

**Keywords:** structural health monitoring, transportation monitoring, pavement monitoring, RF sensing, sensor network

## Abstract

Transportation infrastructure experiences distress due to aging, overuse, and climate changes. To reduce maintenance costs and labor, researchers have developed various structural health monitoring systems. However, the existing systems are designed for short-term monitoring and do not quantify structural parameters. A long-term monitoring system that quantifies structural parameters is needed to improve the quality of monitoring. In this work, a novel Transportation Rf-bAsed Monitoring (TRAM) system is proposed. TRAM is a multi-parameter monitoring system that relies on embeddable backscatter-based, batteryless, and radio-frequency sensors. The system can monitor structural parameters with 3D spatial and temporal information. Laboratory experiments were conducted on a 1D scale to evaluate and examine the sensitivity and reliability of the monitored structural parameters, which are displacement and water content. In contrast to other existing methods, TRAM correlates phase change to the change in concerned parameters, enabling long-term monitoring.

## 1. Introduction

Roads, railroads, and airport runways represent essential critical transportation infrastructure. Aging, massive urbanization, overuse, and climate changes, however, contribute to infrastructure distress, while limited budgets and hidden subsurface faults impair timely maintenance and repairs [1]. These challenges could be addressed by structural health monitoring (SHM) of the subsurface courses (layers) of the infrastructure pavement systems. SHM would include recording and analyzing important parameters of the subsurface with sufficient time periodicity [2]. With this type of SHM, one can constantly assess the evolution of the structural health over time, its current condition and performance, identify potential distressing processes at early stages, and model and predict future distress progression and the corresponding deterioration of the condition and performance of subsurface infrastructure layers [3]. The gained information could then be used to prevent catastrophic failures, trigger and optimize future maintenance activities, and improve safety. This in turn will save financial resources, extend the lifespan and improve the sustainability of the transportation infrastructure, and increase the resilience and sustainability of communities. The existing pavement monitoring techniques focus only on the surface course, the failure of which mostly affects the comfort and safety of drivers but does not lead to structural failure of the pavement and breakdown of the transportation network [4]. In contrast to these existing surface monitoring approaches, the emphasis of the vision proposed here is on a new SHM system for subsurface courses. Subsurface courses have a structural function (i.e., sustaining traffic loads) whose failures are often catastrophic, resulting in the breakdown of the transportation network and loss of life [5]. An early identification of distress in subsurface courses can shed light on the dynamics of distress processes in situ and can significantly improve the way we maintain, operate, and protect transportation infrastructure. More specifically, we describe a vision for a novel approach to address the monitoring of the Transportation System Health, an approach that is integrable into advanced features of future Smart Transportation Systems, such as self-driving, connected vehicles, and infrastructure-assisted driving. We refer here to this new proposed SHM approach as Transportation Rf-bAsed Monitoring (TRAM).

The originality of the TRAM system lies in its proposal, for the first time, of a passive (batteryless) sensing system that allows large-scale, multi-parameter sensing in the large volumes of subsurface courses, with fine-grain spatial resolution. The methodology of the TRAM system for multi-parameter monitoring relies on embeddable backscatter-based, batteryless, radio-frequency sensors (BBRSs) with capabilities to evaluate a multitude of key parameters of the subsurface layers through RF-enabled sensing and with the capacity to carry out some basic data processing. Large numbers of low-cost passive BBRS are embedded in critical segments of the subsurface courses, hence creating a pervasive 3D network that fully covers the volumes of the instrumented segments, and practically enabling the sensing of every point in these segments. The processed data collected from the large volumes of subsurface courses are then transmitted to a remote computer and analyzed by machine learning methods to assess the current, and predict the future, conditions and performance of the courses. Once such an infrastructure is deployed, the system can be reprogrammed to enable the monitoring of additional parameters (e.g., weight and speed of vehicles, and erosion) without the need for hardware upgrades, which in turn will allow the implementation of other future applications with little to no additional costs, thus creating sustainable infrastructures with improved co-benefits and growing value with time.

While the BBRS are based on RF-enabled sensing, the way the RF is used in the proposed system is completely new and transformative: once the BBRS are embedded in the courses, each of them establishes communication channels with all neighboring BBRS. They then monitor the received signals (channels) from neighboring sensors and evaluate a multitude of parameters that provide information about the material between the sensors, such as displacement, strain, cracking, stiffness, humidity, and temperature. This is a paradigm shift from the traditionally used RF sensing tags, where transmitted signals are typically used for communication only. It is also a paradigm shift from traditional sensing approaches, where each parameter requires the deployment of specific and respective types of sensors, while the proposed methodology uses a single type of sensor for monitoring a wide range of parameters [4]. Thus, in the TRAM vision, communication is intimately integrated with sensing. As a result, for the first time, a comprehensive set of key parameters can be monitored throughout large volumes of subsurface material continuum, rather than at a sparse discrete set of points. The sensing will allow for the detection, localization, quantification, and prediction of long-term degradation processes in subsurface courses.

The aim of this paper is to present the vision of TRAM, substantiate its effectiveness, and investigate the physical foundations of its feasibility through experimental study.

## 2. Background Information and Motivation

Pavements are layered structures that provide the following functions: (F1) structural capacity, i.e., strength to support the traffic loads; (F2) durability, i.e., resilience to premature deterioration due to environmental influences; (F3) ride quality, i.e., smooth wearing surface; and (F4) safety, i.e., skid-resistant wearing surface. Referring to Figure 1, the basic components of a typical pavement structure are the following courses: (C1) surface, (C2) base, (C3) sub-base, and (C4) subgrade. Depending on the purpose and design, each of these courses can have different properties and subcomponents (e.g., the base can be unbound, treated, or bound with asphalt, or the sub-base is typically unbound but can be replaced with asphalt or omitted).

Once constructed, pavement is expected to last on the order of decades before it needs replacement. Pavement inevitably deteriorates over time due to a combination of traffic and environmental effects. A warning of such deterioration can be crucial for the prevention of damage escalation, and by contrast, if a warning is absent, it can lead to undetected road distress, which can develop into a failure and result in injuries, loss of life, losses to economy, and the disruption of society. The scope of the SHM system described here is limited to subsurface courses (C2–C4) because they provide structural capacity and long-term durability, i.e., the functions (F1–F2), which are crucial for preventing the breakdown of transportation networks. Although we focus in this paper on the road pavement subsurface, due to the similarities of subsurface materials, the framework described here can be easily extended to other transportation platforms, such as railroads or airport runways.

Based on the manual of the Federal Highway Administration, distress of pavement refers to “conditions that reduce serviceability or indicate structural deterioration” [1]. In the context of the system described in this paper, a failure occurs if “… a portion of the pavement is structurally impaired at any time during the performance period with incipient failure anticipated from the local distress.” [1]. In general, a roadway section can experience distress or failure in numerous ways due to various causes but the major groups of causes are as follows: (D1) geotechnical: moisture/drainage, freeze/thaw, swelling, contamination, erosion, spatial variability, liquefaction, and land-sliding; (D2) construction errors: insufficient base/sub-base strength or stiffness, and poor compaction; and (D3) traffic loads: overweight vehicles. While the consequences of these causes manifest on the roadway surface in the form of cracking, rutting, corrugations, bumps, depressions, potholes, etc. [1], the causes (D1–D3) themselves occur in the subsurface courses. Hence, the multi-parameter monitoring of deep subsurface courses (which is missing in the existing systems) is crucial for the early detection and identification of degradation processes, timely repair, and optimized maintenance.

Recent road failures, due to the given causes (D1–D3), illustrate their adverse impact on society. As an example, some 150 feet of California State Route 1 were washed out in a landslide, and Caltrans estimated the total cost of repairing the roadway and clearing the storm damage to be USD 11.5M [6]. Further, Sun et al. estimated that between 2006 and 2016, damage to roads caused by overweight truck traffic cost Louisiana State USD 17M [7]. An example of the devastating damage and disruption of the transportation network, Figure 2 shows the collapse of Harbor Road in Stony Brook, NY, due to the August 2024 heavy precipitations [8].

Structural damage to pavements happens in subsurface courses and may or may not manifest visibly on the surface [1,9]. Even when visible on the surface, its causes, significance, and future progression cannot be inferred from surface observations, and that is why the continuous or semi-continuous SHM of subsurface courses is crucial. Commercially available methods for assessment of the subsurface are summarized in Table 1 [9]. Common restrictions of these approaches are that each time an assessment of the road is carried out, there is the need to close the road or part of it and bring, deploy, and withdraw measurement instruments [1,9]. While the recent development of specialized monitoring vehicles (e.g., VOTERS [10] and RABIT [11]) mostly focus on surface inspection, they also carry some subsurface sensing devices (see Table 1) that make semi-continuous monitoring possible. However, these vehicles are very expensive and, more importantly, their equipment provides limited information on subsurface structures that is mostly delivered in a qualitative form.

Numerous past efforts aimed at developing methods for comprehensive quantitative continuous or semi-continuous SHM of pavements (e.g., [4]). A common feature of these methods is that they mostly focus on surface course monitoring and not subsurface course monitoring, unlike the system described in this paper. In methods where the subsurface is monitored, it is assumed that the sensors are powered by electrical network or batteries for both the sensing operations and for communications. An overall assessment for these settings is as follows: (1) discrete sensors can measure parameters in their surroundings only, and cannot provide information even for points at modest distance from the sensors’ locations; (2) the wirings of sensors are subject to significant deterioration, especially during the laying down and compaction of the paving materials; (3) the batteries embedded within the wireless sensors severely limit the lifetime of the systems, frequently to a few months only, in deployed conditions; and (4) the sensors’ durability and reliability in the long term are inadequate, initially due to the paving process and later due to the elements (weather), often confining the system lifetime to days or weeks only [4]. Consequently, sensors based on RF signals, as in the system presented in this paper, hold promise to meet these challenges.

## 3. The Design Principles of the TRAM System

The key component of the TRAM system is batteryless sensors (BBRSs) that consist of an RF antenna, low-power electronics to operate the sensors and to process data, and physically strong and chemically passive encapsulation. Many thousands of such miniaturized (centimeter size) sensors will be dispersed (e.g., by burying or mixing them with the materials) within the subsurface courses during the construction of new pavements or repair/reconstruction of existing ones, thereby creating a very dense 3D network of sensors. We do not envision instrumentation of the entire lengths of the roads but only segments of roads which are recognized (identified) as hazardous areas. Exciters, placed aboard maintenance vehicles, will send RF signals to the sensors and power them wirelessly, enabling the actual sensing process and the communication among the sensors. Remarkably, the sensing and measurement operations are based on the changes in the RF signal traversing the subsurface material continuum between the sensors while they communicate with each other. Receivers, placed on the same vehicles, will wirelessly collect the data, send them to a remote central computer to perform detailed data analysis and alert engineers and decision makers about the infrastructure’s condition. Each sensor is equipped with an identification number (RFID). Consequently, the repeatability of exciter and receiver positions relative to the sensors in subsequent measurements is not of importance. As long as two pairs of neighboring sensors receive power from the exciter, they can perform the measurement and report it to the receiver, indicating the measurement location based on their RFIDs. An overall schematic showing the TRAM system is given in Figure 3 along with the parameters that it is supposed to monitor.

To achieve the aim of this approach, fundamental paradigm shifts are required in the design of the system. The BBRSs are powered by an external RF signal and operate at near-zero power regime. They communicate by backscattering the same RF signal that powers them. They possess a unique RF-based sensing ability of measuring the characteristics of the intermediate wireless backscatter channel (i.e., of the signal that travels through the subsurface material). This, in turn, gives the BBRSs the ability to localize themselves and to sense the properties inside the subsurface courses based on the channel characteristics that they “observe”. The measurements from the sensors are analyzed via advanced machine learning methods and can be combined with publicly available environmental satellite data to carry out the identification, diagnosis, and predictive analytics of distress states of the pavements. The proposed sensing system features unprecedented capabilities: for the first time, it will enable advanced subsurface material monitoring at a fine-grain spatial resolution, high fidelity, and at large scales. It will greatly impact our understanding, research, and management of future transportation infrastructures, and it will contribute significantly to the resilience and sustainability of our society and the environment.

RF sensing, based on technology of RF Identification (RFID) tags, provides a unique solution to the above challenges. The tags can be produced in large volumes at an estimated cost of USD 0.10 per tag [12]. Batteryless RFID tags last for decades and can be successfully used in long-term structural health monitoring, as they are powered by an external RF reader [13]. RF-based sensors can be used to monitor various parameters such as relative displacement, strain, cracking, humidity, and temperature [14]. For example, researchers at MIT developed an Amplitude Modifying RFID TABS moisture sensor that successfully functions when the RF reader is at a distance of up to 80 cm [15,16]. Researchers at North Dakota State University demonstrated the viability of ultra-high frequency RFID sensors, which can detect moisture content in soil at levels of 15% by weight or less. The analysis of communication properties between RFID tags and a reader were used to train neural networks for the estimation of volumetric water content [17]. The system was validated in a three-year field study with 9 RFID tags placed 40 cm deep in the soil, which also demonstrated the durability of the sensors. The Federal Highway Administration developed a Smart Pavement Monitoring System that includes RF sensors but only for surface pavement monitoring [18]. And while these examples demonstrate the feasibility of the use of RF-based techniques in subsurface SHM, all of them suffer from two important shortcomings: (1) the need of each sensor to communicate to reader on a one-to-one basis, and (2) the lack of techniques that use sensor-to-sensor communication channel properties to sense the parameters of the 3D measurements throughout the volumes of material continuum. Such techniques could provide a much richer set of measurements, as there are many more sensor-to-sensor links.

The author of [19] developed the backscatter tag-to-tag network, which addressed the second shortcoming mentioned above. The author of [20] introduced a potential application of the passive sensing system. However, the concept deals with a stationary (immobile) exciter and receiver, and the experiments were only preliminary, mostly qualitative; they focused only on displacement monitoring, and they did not provide sufficient analysis to demonstrate the reliability and accuracy of the system. In the transportation monitoring application, a stationary exciter is not ideal due to the limitation of coverage and interruption of traffic. In addition, no experiments on water content detection were performed, which is one of the major failure causes of subsurface courses. The TRAM system proposes solutions for the above challenges, while the controlled quantitative tests presented in this paper demonstrate, for the first time, the system precision in measuring displacements and water content in sand. This confirms that the system can perform multi-parameter monitoring in transportation applications with high precision and supports future development of the TRAM.

### 3.1. Integration with Vehicular Networks

The primary focus of the TRAM system is on the SHM of transportation infrastructures, and to accomplish this, connectivity of the proposed sensor network to the “outside world” is needed. Such connectivity could be integrated within the broader concept of the so-called vehicles-to-everything (V2X). The V2X connectivity opens a wide range of exciting future applications that the proposed TRAM system could support: sensor networks for self-driving vehicles, infrastructure-assisted driving, and connected vehicles.

In general, the concept of connected vehicles (CVs) is a part of the bigger umbrella of Smart Transportation Systems (e.g., see [21,22]). The main idea behind the CV concept is that, using short-range communication among vehicles in close mutual proximity, a much better coordination of operation of such vehicles can be achieved, leading to a reduced rate of accidents and higher transportation safety. The notion of CV is often referred to as vehicle-to-vehicle (V2V) communication. In the broader context, CV will be extended to also include communication between vehicle to infrastructure (V2I) and vehicle to pedestrians (V2P), furthering the goal of safe and reliable transportation systems. Together, such systems are often referred to as V2X systems (e.g., see [23,24]). The purpose of such systems is not only to communicate but also to sense the environment. For example, a V2P system should be able to reliably alert a pedestrian in a timely manner in the vicinity of the vehicle’s path or, reversely, a vehicular driver (or the vehicular hardware in self-driving cars) to the presence of a pedestrian on the vehicle’s path. It is predicted that the CV technology can reduce the number of unimpaired vehicular accidents by 80%. While the study of V2X is out of the scope of this paper, it is important to note that the networks of embedded BBRS, which are part of the proposed TRAM system, can be a key enabler of the V2X concept, as they can be used for the dissemination of V2X information.

### 3.2. Cost–Risk Analysis and Economic Justification

The cost per mile of a new lane of road pavement ranges between USD 2.5M (rural, flat, interstate road) and USD 64.2M (equivalent high cost in major urbanized freeway, expressway, and interstate road [25]), which corresponds to USD 500 to USD 12,000 per foot, respectively. Typical damage due to the structural collapse of a road can lead to multi-million direct material losses, corresponding to several tens of thousands of dollars per foot. However, this does not account for additional losses incurred by the road users and the economy (e.g., the unavailability of the road due to the collapse of the Minneapolis Bridge incurred losses of USD 500,000 per day for road users and the Minnesota economy [26]). In addition, this loss also does not account for the injuries and losses of human lives. In engineering practice, there is no established method for performing the cost–benefit analysis of such long-term monitoring (see [27]). Based on experience, for typical civil structures such as bridges, buildings, pipelines, etc., the cost of monitoring system usually amounts to 0.5–2% of the cost of a new structure [28]. Assuming that BBRSs will be dispersed in subsurface courses at mutual distances of about 6–8 inches, and for a typical depth of subsurface layers of 14 inches, it is expected that 200 to 400 of BBRS devices will be embedded in the subsurface courses of a lane per foot, amounting to USD 20–USD 40 per lane per foot. While this corresponds to 4–6% of the cost of the new pavement system in the case of the least expensive flat rural roads, it represents only 0.15–0.30% in the case of major urban roads, which is comparable with, and even less costly than the applications of the SHM to the above-mentioned types of structures [28]. Given that a failure of a major urban road will adversely impact a significantly larger population and the economy, this cost of monitoring is acceptable and fully justified. If one makes comparisons with potential direct losses in the case of failure, which are counted in tens of thousands of dollars per foot, hundreds of thousands of dollars for road users and economy per day, and human injuries and life losses, the cost of the TRAM system is even more compelling not only for major urban roads but even for small and larger urban roads.

Note that the above considerations assume that every foot of every road lane is instrumented, which would not be the case in real-life settings; rather, in real-life settings, the BBRSs will be strategically embedded only in regions of the road pavements with recognized potentially hazardous geotechnical conditions (e.g., landsliding and flooding). This, in turn, will significantly bring down the cost of monitoring, further favoring the system in cost–risk analysis, while extending the applicability of the TRAM system to mountainous, rolling, and even the least expensive flat rural roads. For example, for 100 miles of long flat rural road, if only critical segments of 5 miles are instrumented, the cost of monitoring will be in the range of 0.2–0.3% of the new pavement system; a similar strategy for major urban roads will decrease the cost to only 0.010–0.015%.

## 4. The TRAM System Design Details

### 4.1. The Monitored Parameters

All distresses (D1–D3) described in Section 2 manifest as 3D deformations, which result from relative displacements (changes in distances) between points in the pavement subsurface courses. Hence, the identification of distresses—i.e., their detections, localizations, and quantifications—directly depends on the capability to detect and localize relative displacements. Therefore, inferring the 3D distribution of these displacements is of paramount importance, as it would directly enable the identification of pavement distress states.

Besides deformations, geotechnical causes of distress (D1) are also characterized by the presence of water in subsurface courses. Consequently, in addition to relative displacements, it is important to monitor the humidity of subsurface courses. A literature review and our results, as discussed below, confirm that BBRSs can be used to monitor both distance (*d*) and humidity (*h*) changes between sensors. Construction and traffic-load causes of distress (D2 and D3) are also related to relative displacements through stiffness, i.e., the relationship between applied loads (due to compaction or traffic, respectively) and deformations. Determining stiffness requires calibration, which can be performed by using traditional approaches such as FWD (see Table 1).

The relative displacement $d_{1}$ between two BBRSs is important for inferring the deformed shape of a pavement system, and it can be derived from *d* as a difference in distance with respect to a reference state $d_{0}$. The average normal strain $d_{2}$ between two BBRSs can be obtained from the ratio $d_{1}/d_{0}$. The cracking $d_{3}$ can be assumed to be equal to $d_{1}$ if the latter is experiencing significant positive changes between two measurements. The strain $d_{2}$ and the cracking $d_{3}$ are important parameters for deducing the performance and closeness to the limit state of courses. The stiffness $d_{4}$ is important for evaluating the pavement’s structural performance, and it can be found from calibration based on the magnitude of applied loads (weight) and the strain $d_{3}$. Given that humidity and displacement *d* are key indicators of pavement distress, and considering that parameters $d_{1}$–$d_{4}$ are all derivatives of *d*, the scope of this paper is limited to a feasibility study on measuring changes in parameters *d* and *h*.

### 4.2. The Novel RF Sensing System

The key to implementation of the TRAM system is an innovative RF-sensing approach integrated in a network of massively deployed (in the hundreds per cubic meter) tiny BBRS devices. They communicate among themselves by the principle of backscattering an ambient RF signal. In the proposed application scenario, the ambient RF signal is generated by dedicated RF exciters, mounted on moving vehicles. BBRS devices carry out the local estimation of the parameters of the RF communication channels with their neighbors, namely, amplitude *A* and phase θ, which allow for the estimation of *d* and *h*, and their derivatives $d_{1}$–$d_{4}$. For granular and long-term monitoring of the observed material, embedded BBRSs have to be integrated into a small-form factor and be self-powered. Although conventional RFID tags provide near-zero power operation in a small-form factor, they require the deployment of costly RFID readers, which limits scalability [29,30,31,32,33,34,35,36,37,38] (see Figure 4a). Additionally, the granularity of the RFID approach is limited by the number of wireless reader-to-tag channels in a centralized system. RF sensors with active radio could provide granularity based on sensor-to-sensor channel estimation, but the power requirement for an active radio would prohibit the self-powered (i.e., batteryless) operation of this type of sensor in the TRAM system.

The proposed solution for the TRAM system represents a paradigm shift to an autonomous system, as the passive RF sensors can communicate with each other without the need for an RFID reader [39,40,41]. If the power level of an ambient RF signal is not high enough in order to establish sensor-to-sensor communication, a continuous wave (CW) signal is generated by a dedicated low-cost device, an exciter, from outside the sensing system as illustrated in Figure 4b. An RF sensor with a receiver based on an envelope detector allows for low-power communication [19,42,43]. The authors of [20,44] devised a technique for estimating the amplitude (*A*) and phase (θ) of a backscatter sensor-to-sensor channel. They demonstrated that the absolute error of θ in the air between RF sensors at a distance of up to 1.5 m is on the order of 15° at a CW frequency of 915 MHz, which is a comparable performance to channel estimation through the use of active radio [44].

This performance translates to a resolution of 5 mm of distance measurements between sensors (based solely on θ).

From the propagation model of the RF wave [45], the unwrapped channel phase is determined as
(1)$$\theta =\pi \frac{d}{v}f-\pi k$$
where $k\in Z^{+}$ to keep θ within the range (0, π), *v* is the velocity of the electromagnetic (EM) wave in the channel medium, and *f* is the frequency of the CW signal. The velocity is a function of the EM parameters (permeability and permittivity) of the material that surrounds the sensors [46]. These parameters are affected by the parameters of the material such as *h*, and the dependence of *A* and θ on each material parameter depends on the frequency of the RF signal. Thus, the use of multiple frequencies for the estimation of *A* and θ enables the simultaneous measurement of multiple parameters *d* and *h*. The selection of the frequencies requires a trade-off between the distance between the sensors, their penetration depth, and the coverage area of the exciter.

### 4.3. Machine Learning for Distress Identification and Predictive Modeling

The network of BBRS will provide large data sets related to the performance and health of the pavement subsurface, in terms of *d*, *h*, and $d_{1}$–$d_{4}$. To extract this information and perform predictions, both physics-based and data-driven models [47,48] based on a machine learning method of choice are needed. One possibility is to work with deep Gaussian processes [49], which are probabilistic models that extend the capabilities of Gaussian processes to multiple layers, analogous to deep neural networks. They are based on Bayesian theory for making inference, and they have the capacity to model highly nonlinear phenomena while, unlike deep neural networks, not being data hungry. The literature shows that both deep learning and Gaussian processes have been used for SHM purposes [50,51,52,53,54,55,56].

We emphasize that the machine learning techniques in the TRAM system are to be deployed mostly in the backbone (e.g., “edge” network), while the operational load on the BBRSs will be mostly limited to acquiring data. This section briefly introduces the planned data analysis approaches to ensure the completeness of the presentation of the vision of TRAM; however, at this early stage of the research (focused on the physical foundations of its feasibility), the development of data analysis algorithms is beyond the scope of this paper.

## 5. Materials and Methods

A series of tests was performed in order to prove the feasibility of the sensing system and to preliminarily characterize its sensitivity and repeatability precision. Hence, the goal of the experimental setup was to simulate small displacements and changes in water content between the sensors embedded in a host material. The authors used a discrete implementation of the RF sensor which interfaced a dipole antenna. The RF sensor implemented a multi-phase modulator and demodulator based on an envelope detector. The multiphase modulator integrated a multi-port RF switch terminated with seven impedances, preselected to provide the phases of the reflection coefficient evenly spaced in a range of 2π. The demodulator implementation included the envelope detector followed by a low-pass filter and a 16-bit 1 MSample/s analog-to-digital converter (ADC).

The prototype experiments were performed in air, dry sand, and wet sand, with two RF sensors. Each sensor was encased inside a dry concrete block to emulate sensor embedding (see Figure 5) and provided the geometrical stability of sensors and repeatability of the tests. The test setups consisted of two sensors: the first was fixed to the base and immobile, and the second was placed on a mobile stage connected to an actuator. The actuator enabled remote-controlled movement of the second sensor. A box (basin) made of Plexiglas was built between the sensors so that it could be filled with sand. A 915 MHz exciter was placed in the middle plane of the box at a distance of 1.18 m with an approximate angle of 44°. Views of the experimental setups for the three performed tests are shown in Figure 6, Figure 7 and Figure 8. To further ensure the repeatability of the tests, they were all performed in a lab at an approximately constant temperature. Therefore, the effect of ambient temperature on the tests was minimized. Nevertheless, while the sensitivity of the system to temperature changes was not rigorously tested due to the lack of equipment, the unintentional temperature changes that occurred during the tests demonstrated that the system is influenced by temperature and will thus require thermal compensation. This is similar to conventional sensors used for monitoring displacement, strain, and cracking, such as sensors based on electrical resistivity, vibrating wires, or fiber Bragg gratings. Quantification of temperature influences on the system and thermal compensation of measurements will be explored in future work.

The first experiment, where the RF signal was propagating through the air, was set as shown in Figure 6. The purpose of this experiment was to verify the behavior of the RF sensors through a comparison of experimental results with the theoretical model. The air was used as the medium because its electromagnetic parameters (e.g., permeability and permittivity) were known and thus, the theoretical model could be derived. The sensor encased in the mobile block was displaced using a linear actuator, while the other sensor, encased in a static block, remained stationary. The linear actuator was calibrated to displace the mobile stage, which was connected to the mobile block, with a change in distance Δ*d* of 1 cm at a time. After each 1 cm displacement, the data were collected for 5 min with a sampling rate of 10 s. The distances between the two RF sensors varied from 89.5 cm to 73.5 cm. An exciter provided the background field to power the sensors. Sensor control, data acquisition, and phase computation were performed on a computer which is not shown in the figures.

In the second experiment, the space between the sensors was filled with dry sand as shown in Figure 7. The purpose of this experiment was to verify the performance of the RF sensors by comparing the test results with an approximate theoretical model when the RF signal propagated through the sand. The distance between the two RF sensors was changed using the actuator and varied from 69.5 cm to 88 cm, with steps Δ*d* of 1 cm at a time. The other test features were the same as in the first experiment.

In the third experiment, the space between the sensors was filled with dry sand, and the distance between the two RF sensors was set to 88 cm and did not vary throughout the test. The purpose of this experiment was to test the capability of the RF sensors to detect the presence of water in the sand. In other words, the RF sensors were tested for their ability to detect changes in water content. Water was added four times, with 200 mL each time. The experiment procedure was as follows: Initially, 200 mL of water was added 11 cm away from one of the sensors. Subsequently, another 200 mL of water was added 22 cm away from the first. Then, the third 200 mL of water was added 22 cm away from the second, and the fourth 200 mL of water was added 22 cm from the third (and 11 cm from the other sensor). The volume of sand under the test was 2.54 × 2.54 × 8.80 = 56.77 cm^3^. Each 200 mL increment represented a volumetric increase of water content by 0.2/56.77 × 100 = 0.35% per step, totaling 1.40% for all four steps. Although the diffusion process of the water into the sand was uncontrollable, the distances between the locations where water was added were set as far apart as possible to avoid the water diffusing to the same location. After each 200 mL of water was added to the sand, the data were collected for 5 min with a sampling rate of 10 s. The experiment setup after water was added is shown in Figure 8.

In each step of each experiment, the exciter would wirelessly send the power to the two sensors, and the two sensors would establish a communication channel. The phase θ of this channel was measured, and its change, Δθ, from the initial (reference) value was evaluated. For the first and second experiments, the function of theoretical change in the phase Δθ, in degrees, with a change in distance Δ*d* is given as
(2)$$\Delta \theta =360^{\circ }\frac{\Delta d}{\lambda }$$
where λ is the electromagnetic wavelength in the observed medium (air or sand). For air, $\lambda_{\mathrm{air}}$ ≈ 328 mm, and for sand, $\lambda_{\mathrm{sand}}\in$ [164 mm, 232 mm] (based on the research studies [57,58]). The value for $\lambda_{\mathrm{sand}}$ has a relatively large range because it varies based on the density and compaction level of sand. The denser and more compact the sand is, the smaller $\lambda_{\mathrm{sand}}$ is.

## 6. Results

Figure 9 and Figure 10 show the results of the first and the second experiments, respectively. The graphs show measured ($\hat{\Delta \theta }$) and theoretical (Δθ) values of the phase shift versus the change in displacement Δd. For the first experiment, which was conducted in the air as the medium, steps of 1 cm were used to assess the system sensitivity and repeatability precision, and to compare the sensor performance with the theoretical values. As shown in Figure 9, the measured value of the changing phase shift over a changing distance, i.e., the system sensitivity, was 12.18°/mm, i.e., $\hat{\Delta \theta /\Delta d}$ = 12.18°/mm, and the theoretical value of the changing phase shift over a changing distance was 10.99°/mm, i.e., Δθ/Δd = 10.99°/mm. The difference between those two values was approximately 10%, which showed, in general, good agreement between the experimental and theoretical sensitivity of the sensing system. The repeatability precision, calculated as the standard deviation of the phase measurement at constant displacement, was 1.33°. The average phase shift with the 1 cm displacement was 13.15°, and the standard deviation of phase shift with 1 cm displacement was 4.9°. Therefore, the system had 95% confidence in detecting 1 cm displacement in the air.

For the second experiment, conducted using sand as the medium, steps of 1 cm were also used to assess the sensitivity and repeatability precision. As shown in Figure 10, the measured sensitivity was 15.18°/mm, i.e., $\hat{\Delta \theta /\Delta d}$ = 15.18°/mm, and the theoretical sensitivity, i.e., Δθ/Δd, ranged from 15.54°/mm to 21.98°/mm. The difference between the measured value and the lower bound of the theoretical value was less than 3%. The standard deviation of the phase measurement at constant displacement was 1.24°. The average phase shift with a 1 cm displacement was 14.68°, with a standard deviation of 6.5°. Therefore, the system had a 68% confidence level in detecting 1 cm displacement in the sand medium. Removing data points with odd number displacement adjusted the statistical calculation to evaluate detection at 2 cm displacement. The average phase shift with 2 cm displacement was 30.42°, with a standard deviation of 12.85°. Therefore, the system achieved a 95% confidence level in detecting 2 cm displacement in the sand.

In the third experiment, conducted by adding water to the dry sand medium, four increments of 200 mL of water were added to assess the system’s sensitivity to the presence of water. As shown in Figure 11, the standard deviation of the phase measurement with unchanged water content was 0.64°. The average phase shift due to the addition of 200 mL of water was 29.2°, with a standard deviation of 7.8° at a constant water content. Therefore, the system achieved a 99% confidence level in detecting the addition of 200 mL (equivalent to a volumetric change of 0.35% in water content).

## 7. Discussion

The first experiment was the only one that could directly compare the measured and theoretical sensitivity to evaluate the performance of the sensing system. The reason was that the second experiment’s theoretical model provided a range of theoretical values. Therefore, the results of the first experiment provided evidence of the sensing system’s performance, compared to the theoretical model. Although the difference between the measured and theoretical values was approximately 10%, this difference was acceptable. Since the experiment was conducted in an indoor setup, the backscatter RF signals from sensors experienced multi-path effects, where parts of the RF signals were reflected by obstacles and superimposed on the measuring RF signals. The multi-path effects increased the noise in the measured signal and affected the measuring properties of the sensing system. Therefore, with the presence of a multi-path effect, an approximate difference of 10% between measured and theoretical values was considered acceptable and sufficient to verify the measurement performance of the sensing system.

In the second experiment, the theoretical sensitivity of sand was affected by its density and compaction. Since no compaction was performed during the second experiment, the experimental results were expected to align closer to the lower bound of the theoretical value, which was indeed the case. Furthermore, conducting the experiment in an indoor setup exposed the system to a multi-path effect as discussed earlier. Therefore, a 3% difference between the measured and theoretical lower bound was considered acceptable. Evaluation of confidence in the displacement detection showed a good confidence level for detecting 1 cm displacement (68%) and excellent confidence for detecting 2 cm displacement. With future improvements in hardware and algorithms, the confidence level of detecting 1 cm displacement in the sand medium could further be improved. The centimeter-scale displacement sensing system proved adequate for long-term monitoring applications in transportation systems.

In the third experiment, the theoretical value was not calculated due to the complex diffusion pathway of water into the dry sand, which could not be accurately determined. This experiment simulated a scenario resembling water infiltration in the sub-base layer of the pavement. The results demonstrated the sensing system’s capability to detect water infiltration in pavement sub-bases.

## 8. Conclusions

This paper identifies the needs for and benefits of continuous or semi-continuous SHM of large volumes of pavement subsurface courses. Currently, no methodology addresses these needs in practice. Therefore, a vision for a new monitoring system, called TRAM, based on embedded, wireless, and batteryless RF sensors (BBRSs), has been developed and presented. Cost analysis demonstrates that the proposed TRAM system holds the potential of providing a highly beneficial economical solution especially for, but not limited to, roads in urban areas. The system uses communication channels established between the passive BBRS embedded in structural subsurface courses of the pavement to sense important parameters that indicate the courses’ structural health and performance. This innovation eliminates the need for the permanent presence of RF readers and distinguishes the TRAM and BBRSs from traditional RF applications, resulting in a paradigm shift to an autonomous system with high 3D spatial resolution. A series of experiments were performed to carry out a feasibility study and preliminarily characterize the sensing system. The experimental results demonstrate the capability of the RF-based sensing system to monitor displacements and detect water infiltration in sand, with high sensitivity and repeatability precision. This capability is transferable to subsurface pavement courses monitoring, and it justifies the creation of a novel, paradigm-shifting, RF-based structural health monitoring methodology—TRAM—for the continuous and semi-continuous assessment of the condition and performance of subsurface pavement courses. An added value of the new methodology is its potential to support future Smart Transportation Systems. Future work will explore new hardware to reduce the influence of multi-path reflections and improve the linearity of the BBRS response to changes in displacement. Furthermore, it will develop data analysis algorithms for the self-localization of BBRSs and the identification and prediction of unusual structural behaviors, the networking of dense arrays of embedded sensors for optimal power and communication management, and large-scale lab and field implementations.

## Figures and Tables

**Figure 1 sensors-24-06018-f001:**
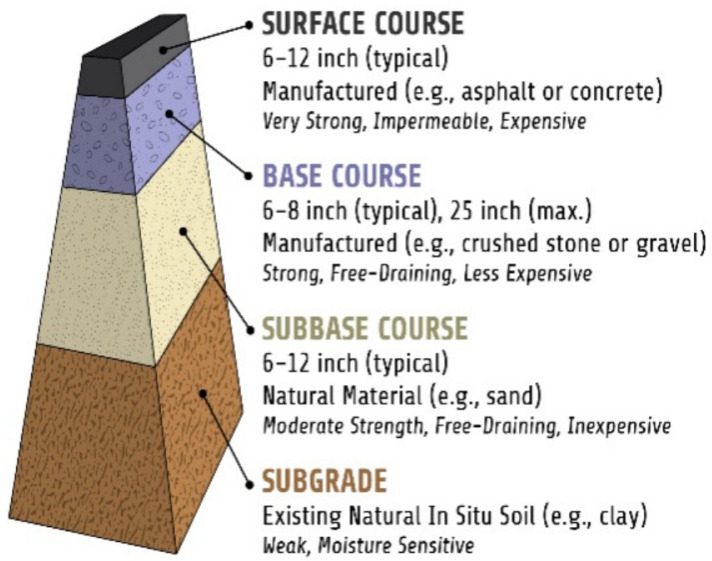
Typical courses of a road pavement system.

**Figure 2 sensors-24-06018-f002:**
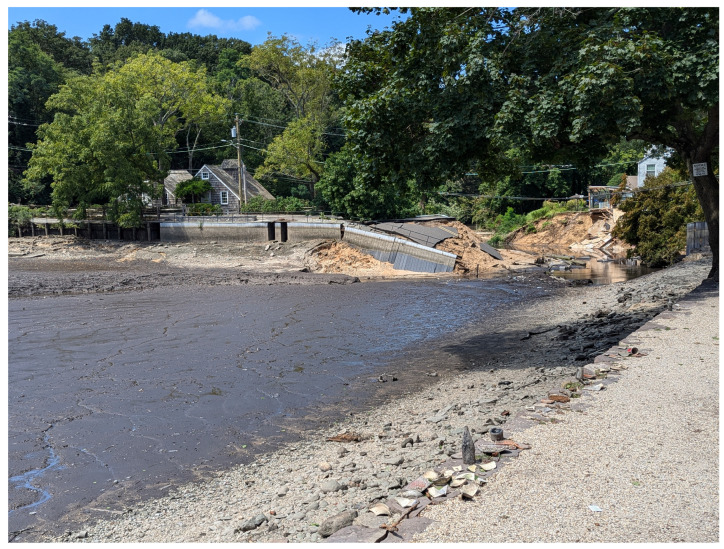
View to the site of the collapsed Harbor Road in Stony Brook.

**Figure 3 sensors-24-06018-f003:**
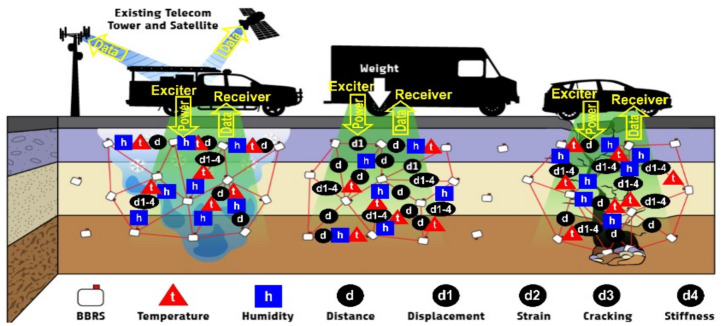
BBRS channel-to-channel communication parameters (amplitude and phase, presented with red lines) are used to measure a multitude of parameters (*t*, *h*, *d*, $d_{1}$–$d_{4}$) in the material continuum between them; they are powered and read from a car in motion, and their measurements are transmitted to a remote computer for analysis.

**Figure 4 sensors-24-06018-f004:**
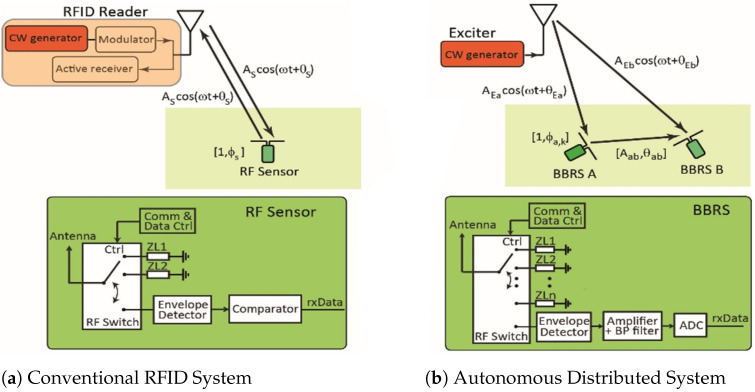
Comparison of (**a**) a conventional RFID system and (**b**) an autonomous distributed system, where sensors a and b communicate with each other in the presence of a dedicated exciter. The channel amplitude and phase are denoted by $A_{ab}$ and $\theta_{ab}$.

**Figure 5 sensors-24-06018-f005:**
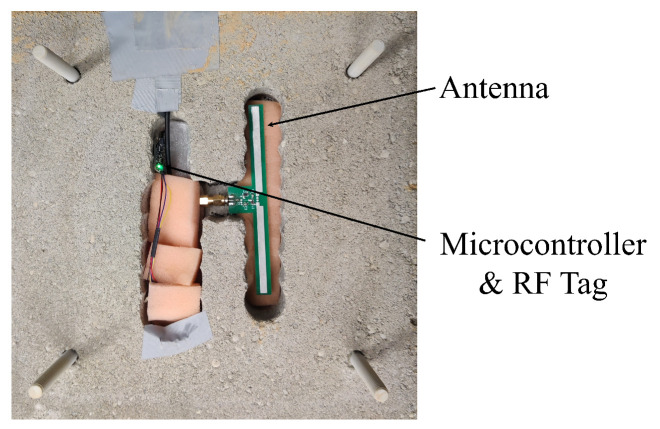
RF sensor, with tag and antenna highlighted, placed inside a milled dry concrete block.

**Figure 6 sensors-24-06018-f006:**
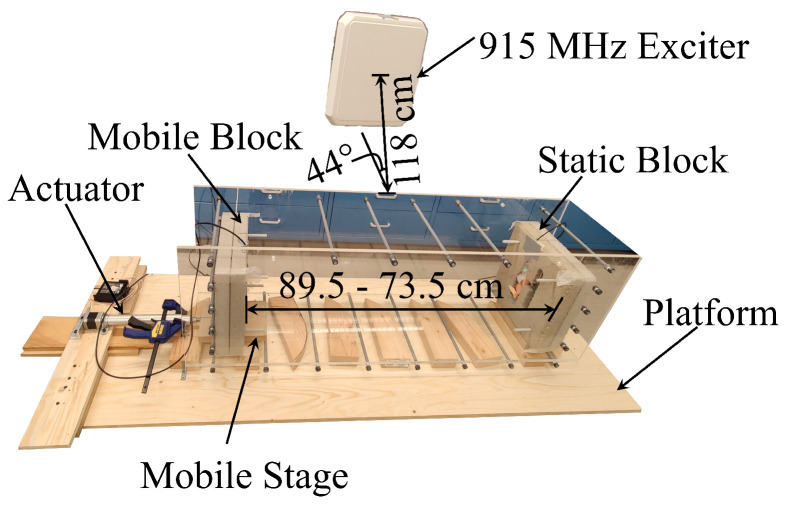
Experimental setup for air medium.

**Figure 7 sensors-24-06018-f007:**
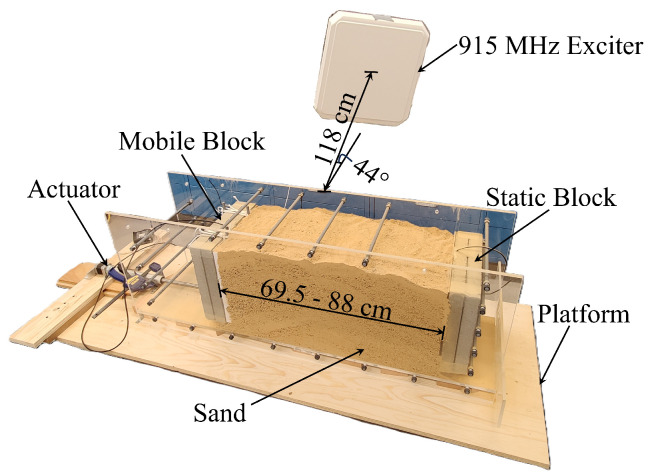
Experimental setup for the second test performed with sand.

**Figure 8 sensors-24-06018-f008:**
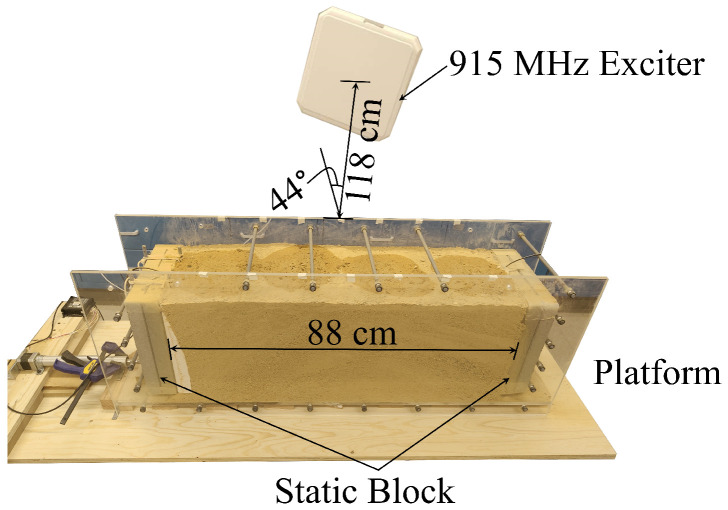
Experimental setup for the third test performed to identify presence of water in sand (setup after the water was added).

**Figure 9 sensors-24-06018-f009:**
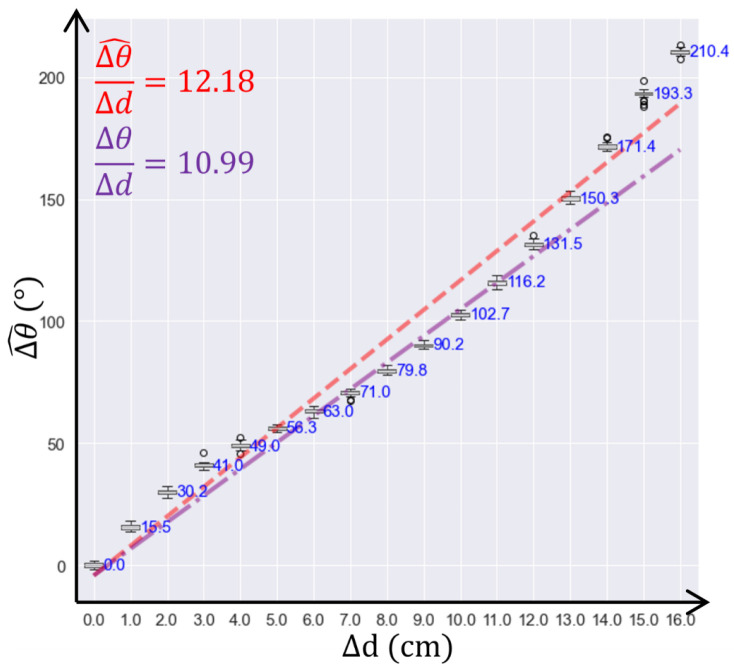
Results of the first test performed in the air (in red); theoretical value (in purple) is given for comparison.

**Figure 10 sensors-24-06018-f010:**
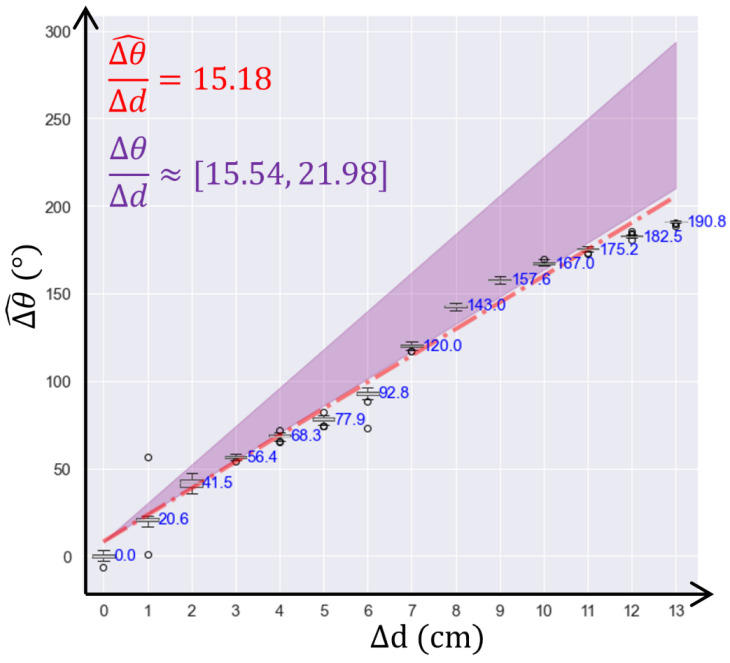
Results of the second test performed in the sand (in red); theoretical value (in purple) is given for comparison.

**Figure 11 sensors-24-06018-f011:**
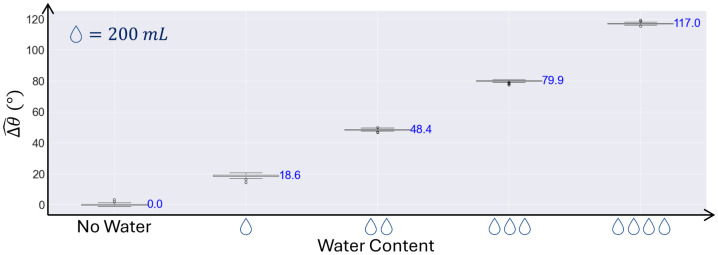
Phase shift after adding increments of 200 mL of water to dry sand; water was added at equidistantly spaced locations to prevent overlapping diffusion.

**Table 1 sensors-24-06018-t001:** Commercially available methods for the nondestructive testing of road pavements [9].

Sensing Principle	Monitored Parameter	Main Limitations
Falling Weight Deflectometer (FWD)	Stiffness of individual layers in a multi-layered system	Closure of road or lane, slow deployment, performed infrequently
Spectral Analysis of Surface Waves (SASW) *	Stiffness and layer thickness	Closure of road or lane, monitoring single structural parameter (stiffness)
Thermal infrared photography	Feature detection (delamination, voids)	Closure of road or lane, limited penetration, no quantification of structural parameters
Ground Penetrating Radar (GPR) *	Layer thickness, dielectric constants, feature detection	Closure of road or lane, limited or no quantification of structural parameters

* as a standalone instrument or as a part of integrated monitoring system on specialized vehicles, such as VOTERS [10] and RABIT [11]); only VOTERS does not require closure of road or lane.

## Data Availability

The data presented in this study are available on request from the corresponding author. The data are not publicly available due to privacy.

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
