# Peer review of "A Vision and Proof of Concept for New Approach to Monitoring for Safer Future Smart Transportation Systems"

_sensors, 2024, doi:10.3390/s24186018_

Round 1

Reviewer 1 Report

Comments and Suggestions for Authors

This paper presents a model for periodic monitoring of pavement structure quality using passive RFID sensors embedded in the pavement system. This approach prevents critical degradation of the road infrastructure through early detection of pavement distress (in situ). It is believed that the insertion of multiple such sensors directly into the material layers that make up the pavement system will allow early detection and identification of degradation processes for repair and maintenance.

In the Abstract (lines 5-7), the authors state: “In this work, a novel Transportation Rf-bAsed Monitoring (TRAM) system is proposed. TRAM is a multi-parameter monitoring system that relies on embeddable backscatter-based, batteryless, and radio-frequency sensors.” However, several key elements presented in this paper have been previously published by some of the co-authors. Of these, the following references exhibit the greatest degree of similarity:

·        [35], published in 2018, where the backscattering tag-to-tag network (BTTN) concept, quite similar to TRAM, is implemented, and

·        [39], published in 2021, shows that (i) a multiphase modulator was implemented within the RF tag, and (ii) Figure 2 shows the utilization of passive estimation of the amplitude and phase of sensor-to-sensor channel, similar to Figure 4b of this paper.

Therefore, the authors are asked to describe in detail the similarities and differences in structure and methodology in [35] and [39] in the revised version of the paper in order to clarify this issue.

The experimental simulations conducted to ascertain the sensitivity of a system comprising two elements, one of which is capable of displacement relative to the other, facilitate the extraction of valuable insights for the subsequent advancement of the sensor system. In a practical scenario, however, the sensors integrated into the pavement structure will occupy arbitrary relative positions with respect to each other due to the deployment technologies.

It can be reasonably deduced that a certain margin exists within which the amplitude and phase of the signal resulting from the combinations of the individual signals will be situated. Furthermore, it seems plausible that the trajectories traversed at each EXCITER pass over the area in which the sensors are embedded do not overlap with those of the previous passes.

What is the estimated effect of these deviations from the previous trajectories, and what measures have been proposed to address them?

What would be the acceptance/rejection criteria for data acquired under these conditions?

Additionally, what is the impact of ambient temperature change?

Reviewer 2 Report

Comments and Suggestions for Authors

This paper presents a method using TRAM system to evaluate the defect or decay of pavement system. The background and foundamentals of the research is presented in thorough, but the novelty of the research is not appealing to reader. In the principle part, the auther mentioned critical sensor design, passive tag communication, RF sensor network, multiple parameter and machine learning etc. However, only phase characteristics is invesitgated for the TRAM sensing system. This makes this paper more like a review paper. It is suggested to add more technique details of the investigation before considering publication.

Reviewer 3 Report

Comments and Suggestions for Authors

sensors-3164120 entitled “A Vision for New Approach to Monitoring for Safer Future Smart Transportation Systems” has been reviewed. The author proposes a novel Transportation Rf-based Monitoring (TRAM) system, which relies on embeddable batteryless backscatter-based radio-frequency sensors. This system monitors structural parameters using 3D spatial and temporal information, enabling comprehensive monitoring of key parameters within large-scale subsurface material continuums. The feasibility of the proposed monitoring system's physical foundation is explored through experimental research.

In the opinion of the reviewer, the study is quite interesting, clear expression and exhibits a certain degree of innovation. The designed experiments are valuable, the research results are credible, and the research content has promising application prospects. The manuscript is suitable for publication in the journal 'Sensors'. However, before accepting this article, several recommended minor revisions or questions should be conducted:

1. The title of the paper should not merely emphasize that it is a perspective or outlook. It is a study with a forward-looking nature, and the authors have conducted some research work on it. It is recommended to revise the title to better align with the abstract and the content of the paper.

2. It is recommended to combine Section 1, "Introduction," and Section 2, "Background Information and Motivation," into a single section. Additionally, it is suggested to strengthen the citations and discussion of relevant literature related to the research background in the paper, rather than having the introduction section without any references or relevant discussions (The introduction of this study lacks sufficient discussion of relevant literature).

3.  All the verbs that express the research done by the authors should be expressed in the past tense.

4. It is recommended to standardize the font style of all figure labels throughout the paper and to ensure that the quality of the figures is clearer and more legible.

Round 2

Reviewer 1 Report

Comments and Suggestions for Authors

The paper is well written and of interest to the field of road transportation. However, the arguments presented by the authors to support the originality of the content in comparison to their previous work are not strong enough, which is why the rating of "Low" for the "Originality/Novelty" criterion is maintained.

Changing the paper type to PERSPECTIVE is accepted.

Reviewer 2 Report

Comments and Suggestions for Authors

This paper has presented a new idea for subsurface monitoring of pavement degradation. Commercial methods mostly focus on surface monitoring while researches have limitation of short-term operation. The proposed system use passive tags to accommandate the challenges in sensing, communication and energy. An initial investigation is undertaken with early stage results proving the concept.

As the authors have changed the category of this paper from general article to Perspective study, it is recommended to include more literature on passive RF researches on transportation systems. To claim the system is new, it would be more convincing to compare the concept with other researches rather than commercial approaches, which have been validated for decades.

In the presentation part, the illustration is clear and the results are presented and discussed in details. I would recommend a minor revision with more references added.
